# Parental Reports of Children’s Dental Pain Experience and Associated Factors among Brazilian Children

**DOI:** 10.3390/ijerph21050599

**Published:** 2024-05-07

**Authors:** Jéssica Aparecida Silva-Rabelo, Letícia Fernanda Moreira-Santos, Júnia Maria Serra-Negra, Cristiane Baccin Bendo, Saul Martins Paiva, Isabela Almeida Pordeus

**Affiliations:** Departament of Pediatric Dentistry, School of Dentistry, Universidade Federal de Minas Gerais, Belo Horizonte 31270-901, Minas Gerais, Brazil; jessicasilva@ufmg.br (J.A.S.-R.); leticiamoreirasantos@ufmg.br (L.F.M.-S.); juniamcsn@ufmg.br (J.M.S.-N.); crysbendo@ufmg.br (C.B.B.); isabelapordeus@ufmg.br (I.A.P.)

**Keywords:** dental caries, toothache, pain, preschool children, pediatric dentistry

## Abstract

Exploring children’s dental pain experiences helps to develop healthcare policies for improving oral health and quality of life. A cross-sectional study involved 300 parents/caregivers of four- to seven-year-old children using snowball sampling. Parents/caregivers self-completed an online questionnaire on sociodemographic characteristics, parenting styles, their child’s oral hygiene practices, free sugar consumption, and dental history. The questionnaire was created using Google Forms and was disseminated to parents/caregivers via E-mail and/or WhatsApp©. Descriptive and Poisson regression analyses were performed (*p* < 0.05). Children’s dental pain experience was reported by 20.3% of the parents. The authoritative parenting style was predominant. The child’s mean age at the first consumption of sugar was 1.38 (±0.64) years, and 40.3% of the children had high-free sugar consumption. The mean age for the first dental appointment was 2.26 (±1.31) years, and 24.3% of the children never went to a dental appointment. The prevalence of dental pain experience was higher in children who attended their first dental appointment later (PR: 1.02; CI 95%: 1.01–1.03) and among those with high-free sugar consumption (PR: 1.90; CI 95%: 1.21–3.00). High sugar consumption and delay in the first dental appointment may increase the likelihood that children will experience dental pain.

## 1. Introduction

Dental pain is a common symptom of oral diseases, classified as a dental emergency that negatively impacts individuals’ quality of life [1,2,3]. Its prevalence varies between different studies and age groups [4,5,6]. Among children, dental pain has a high prevalence [4,5]. Within Brazilian preschool-age children, its prevalence seems to range from 9.4 to 21.1 percent, and it is often associated with dental caries [1,7]. In this context, the experience of dental pain has been considered the most common symptom or consequence of the presence of untreated dental caries [8].

Dental caries is the most prevalent oral disease in the world. In Brazil, the most recent national epidemiological survey reports that the prevalence of early childhood caries (ECC) in five-year-old children is 53.4 percent [9,10]. According to the American Academy of Pediatric Dentistry, ECC is defined as the presence of one or more decayed (non-cavitated or cavitated lesions), missing (due to caries), or filled surfaces in any primary tooth of a child under the age of six [11]. The consequences of dental caries in children’s lives include pain, decreased appetite, difficulty chewing, weight loss, difficulty sleeping, poor school performance, hospitalizations, emergency care, high treatment costs, and a negative impact on oral health-related quality of life [2]. Biological, behavioral, and psychosocial risk factors are associated with ECC development [12,13]. Although it is highly prevalent, ECC is a preventable disease since almost all of its risk factors are modifiable and can be prevented through measures to promote oral health implementation [12,13].

Global organizations have developed recommendations in an attempt to reduce the prevalence and impact of ECC in the world, such as the child’s first dental appointment being in the baby’s first year of life when parents/caregivers have access to oral health information [11,13,14]. Among this information, the following can be listed: brushing the teeth with an adequate amount of fluoride toothpaste (1000 ppm) twice a day, controlling free sugar consumption, avoiding the child’s consumption of sugar before two years of age, and limiting it after this age [11,13,14]. 

As known, these practices to ensure a child’s oral hygiene and dietary habits are influenced by parental attitudes that may have a direct impact on children’s oral health [15,16]. Baumrind (1971) defined that the set of attitudes and practices that parents experience with their children comprise parenting styles [17]. Through their educational practices, parents transmit to their children’s values that they consider important. Three main parenting styles were described: authoritarian, authoritative, and permissive. The authoritarian parenting style is characterized by manifestations of power and low levels of support. The authoritative parenting style is characterized by having firm standards of controlling the child’s behavior and high levels of demand; however, parents tend to be affectionate and supportive. The permissive parenting style is characterized by acceptance of the child’s wishes, actions, and impulses [17]. 

Some studies demonstrate that parenting style can influence children’s general health since associations were found between it and children’s obesity and emotional issues [15,16,18]. In this sense, studying the possible association between parenting styles and oral health outcomes in childhood, such as dental pain experience, seems relevant given that investigations on this topic are limited in pediatric dentistry research [16,19]. Understanding the variables that affect children’s behavior can contribute to preventing oral health problems [9,10]. 

The present study aimed to assess the prevalence of the parental reports of children’s dental pain experience and associated factors, such as sociodemographic factors, parenting styles (more or less authoritative, more or less authoritarian, and more or less permissive), free sugar consumption, and oral hygiene practices in Brazilian children. It was hypothesized that children with high sugar consumption and whose parents had more authoritarian and permissive attitudes would have a higher prevalence of dental pain experience.

## 2. Material and Methods

### 2.1. Study Design and Sample 

A cross-sectional online study was carried out in Divinópolis, which is located in the state of Minas Gerais in the southeast region of Brazil. The city has a Gini Index of 0.47. The study sample consisted of parents/caregivers of children aged four to seven years of both sexes living in Divinópolis. The study was conducted at public and private preschools and elementary schools. Preschool covers the age range from 4 to 5 years old. There are 74 preschools in Divinópolis (37 public and 37 private). Elementary education includes children aged six to fourteen, and the city has 73 schools (51 public and 22 private). The study population was recruited using snowball sampling [20]. In this method, the first subject is recruited, and then he/she provides multiple referrals. Each new referral then provides more data for referral and so on, until saturation. In this study, the saturation point was determined based on the data collection period (between August and November 2021). The first referrals were parents/caregivers of children from 10 schools (five public and five private) selected for convenience. Additionally, a group of health professionals (pediatric dentist, pediatrician, nutritionist, speech therapist, and psychologist) and parents/caregivers were invited to assist in the questionnaire dissemination. Parents/caregivers of children with neurological disorders, such as autism and epilepsy, in addition to systemic diseases, such as diabetes, were excluded.

### 2.2. Data Collection

Data collection was performed based on an online and self-administered questionnaire answered by parents/caregivers. The questionnaire was created using Google Forms (Google Inc., Menlo Park, CA, USA) and was disseminated to parents/caregivers via E-mail and/or WhatsApp© (WhatsApp Inc., Menlo Park, CA, USA). The time to complete the questionnaire was estimated to be around 10 to 15 min, and after submission, responses could not be revised or changed.

### 2.3. Questionnaire

Parents/caregivers completed a six-page online questionnaire. The first page contained the consent form. After agreeing to participate, parents/caregivers were directed to the next page, which addressed sociodemographic characteristics, including the child’s sex and age, type of school (public or private), family income (using the Brazilian monthly minimum wage [BMMW] as a reference, which was approximate to USD 215, dichotomized by the median, as ≤ or > three times the BMMW) and parents/caregivers schooling (dichotomized by median as ≤ or >12 years of study). In addition, questions about the diagnosis of neurological and systemic diseases were included.

The third page of the questionnaire contained a dietary questionnaire regarding free sugar consumption and the child’s age at their first consumption of sugar [21]. Free sugars included all mono- and disaccharides added to foods and drinks by manufacturers, cooks, or consumers, plus the sugars that are naturally present in honey, syrups, fruit juice, and fruit juice concentrates. Free sugars do not include sugars naturally present in milk and milk products or whole fresh fruits and vegetables [22]. The questionnaire contained four items as follows:How often does your child drink sugar milk or chocolate milk?How often does your child drink soft drinks or juice boxes?How often does your child eat candies, lollipops, and chocolate?How often does your child eat stuffed cookies?

Parents/caregivers were asked to respond to these items based on a six-month recall. The response options included “never or rarely”, “once or twice a week”, “three to six times a week”, “once or twice a day”, and “three or more times a day”.

The fourth page of the questionnaire contained data regarding oral hygiene practices, including the child’s toothbrushing frequency (dichotomized into two or more times a day and none or once a day) and the use of fluoridated toothpaste (“yes” or “no”) [23].

The fifth page of the questionnaire investigated the parental reports of children’s dental pain experience through the following question: “Has your child ever felt dental pain?” [4]. In the present study, this question was directed to parents/caregivers, who were asked to answer according to the answer options of “yes” or “no”. In addition, it was investigated if the child had already been to the dentist and the age at their first dental appointment.

The sixth page of the questionnaire assessed the parents’/caregivers’ parenting styles through the Parenting Styles and Dimensions Questionnaire (PSDQ) [24]. This instrument was adapted and validated for the Brazilian culture and has 32 questions addressing parenting styles through three domains: authoritative, authoritarian, and permissive. Parents/caregivers were asked to answer questions about their attitudes toward child behavior based on a scale that ranged from never (0) to always (5). The number of items for each domain was as follows: authoritative parenting domain, 15 items; authoritarian domain, 12 items, and permissive domain, 5 items. The mean of each parenting style domain was calculated, and the highest average score among the three domains indicated the predominant parenting style [25]. Cronbach’s alpha index of the instrument was evaluated, obtaining a satisfactory value of 0.74.

### 2.4. Pilot Study

A pilot study was carried out with ten parents/caregivers to test the applicability of the proposed methodology. Pilot study participants were selected for convenience and were not included in the main study sample. No change in methodology was considered necessary.

### 2.5. Statistical Analysis

The results were analyzed using SPSS software (SPSS for Windows, version 21.0, IBM, Armonk, NY, USA). The two-step cluster analysis, using the log-likelihood distance measure, was performed to define the high and low-free sugar consumption groups according to the responses to the four items of the dietary questionnaire. Two-step cluster analysis was used to consider the pattern of responses for each item separately and the importance of each item to the formation of clusters. The Chi-square test was used to demonstrate the pattern of responses for each item. 

In the present study, the parenting style was incorporated into the analysis through the mean scores (quantitative variables) of the three domains of parenting styles. This analysis was valid because parenting styles were conceptualized as non-mutually exclusive typologies. Thus, although a parent/caregiver had a dominant parenting style, they may present more or fewer attitudes related to each of the three domains. Descriptive statistics were performed to characterize the sample. In addition, bivariate analysis was performed through the Chi-square test and Mann–Whitney test (*p* < 0.05). Finally, the multiple analysis was performed using Poisson regression with robust variance. Variables with *p* < 0.20 in the bivariate analysis were incorporated into the adjusted regression model. Those with *p* < 0.05 in the final model were considered statistically associated with the parental reports of the children’s dental pain experience.

## 3. Results

A total of 315 questionnaires were collected. Seven parents/caregivers were excluded because their children did not meet the age group criteria, and eight were excluded due to their children being diagnosed with autism, epilepsy, or acute disseminated encephalomyelitis. Thus, the final sample consisted of 300 parents/caregivers. The mean age of the children was 5.45 years (±1.16), and that of parents/caregivers was 35.0 years (±6.10). 

Table 1 displays the characterization of the sample. The prevalence of dental pain was 20.3%. Most children were female (50.3%), used fluoridated toothpaste (91.7%), brushed their teeth two or more times a day (90.7%), and had already been to a dental appointment (75.7%). The mean age for the first dental appointment was 2.26 years (±1.31). Among the parents/caregivers, 55.3% reported >12 years of study, and 52.0% reported a monthly family income of ≤three BMMW. The authoritative parenting style was predominant (98.6%).

Table 2 shows that the frequency of the four items of the dietary questionnaire was statistically higher for the high-free sugar consumption group than for the low-free sugar consumption group (*p* < 0.001), demonstrating that there were genuine differences between the groups. 

The results of the bivariate analysis are shown in Table 3 and Table 4. The following variables were associated with the parental reports of children’s dental pain experience (*p* < 0.05): parents’/caregivers’ schooling, free sugar consumption, the child’s age at the first dental appointment, the child’s age at the first consumption of sugar, and the authoritarian parenting style domain. 

The Poisson regression model is shown in Table 5. The final model demonstrates that a one-year increase in the age of the first dental appointment was reflected by a two percent increase in the prevalence of parental reports of children’s dental pain experience (RR = 1.02; 95% CI = 1.01–1.03; *p* < 0.001). In addition, children in the sample with high-free sugar consumption had a 1.9 times higher prevalence of experiencing dental pain, according to the reports of parents/caregivers (RR = 1.90; 95% CI = 1.21–3.00; *p* = 0.006).

## 4. Discussion

In the present study, the prevalence of parental reports of children’s dental pain experience was 20.3 percent. Similar results were found in a study that investigated dental pain among Brazilian preschoolers, and higher prevalence rates were reported in investigations involving older children [4,7,10,26]. Comparisons between these studies should be made with caution because of the differences in samples and research methods used. However, the high prevalence of dental pain experience observed may be due to a common factor: the high prevalence of dental caries in Brazilian children [10]. Although dental caries is a preventable disease [12,13], in Brazil, as in other developing countries, its prevalence is still high [9,10,12,13]. 

The hypothesis that parental styles could be related to the child’s dental pain experience was not confirmed. Some investigations have found an association between parenting styles and children’s health outcomes, such as dietary habits, obesity, and dental caries [15,16]. In the literature, the authoritative parenting style is usually associated with healthier behaviors, and children of permissive parents/caregivers show a higher dental caries status [27]. It is also important to note that an indirect relationship between parents’/caregivers’ attitudes and the child’s dental pain experience was observed in this study. The significant association found between high-free sugar consumption and the prevalence of dental pain experience reveals that parents’/caregivers’ attitudes may be reflected by dietary habits. This finding can be explained considering that dental caries is the main reason for experiencing dental pain and has a multifactorial etiology, including characteristics of parents/caregivers, such as sociodemographic conditions, health beliefs, locus of control, self-efficacy, and literacy in oral health. These characteristics may influence parents’ knowledge about health behaviors, such as dietary habits adopted by their children [14,28]. High-free sugar consumption is strongly associated with the development of dental caries, in addition to weight gain and obesity [29,30]. Thus, the consumption of free sugars is of critical importance to the development of dental caries and, consequently, for the report of the child’s dental pain experience [31,32]. Efforts should be made to guide parents/caregivers to control the consumption of free sugar in children, avoiding its consumption before two years of age and limiting it after this age [13,14].

In addition, the first preventive dental appointment is considered fundamental for promoting oral health during childhood [11]. Parents/caregivers should take their children to the dental office when the first tooth erupts or up to one year of age [11]. Delaying the child’s first dental visit is usually associated with future dental emergency appointments and restorative procedures [33]. In this investigation, delays in the first dental appointment were associated with the parental reports of children’s dental pain experience. The early initiation of dental care ensures the opportunity to provide preventive dental care through oral health counseling once parents have the opportunity to receive important information about oral health care, including free sugar consumption control [11,13,14]. 

Regarding socioeconomic characteristics, several studies have established associations between dental pain experience and caregiver’s schooling and family income, whilst children of parents with low educational levels and lower socioeconomic status tend to experience more dental caries and, consequently, more dental pain [3,4]. High parental educational levels may also indicate a greater chance of developing healthy habits related to oral health and making healthier choices in their diets [3,4]. Although the present investigation did not find such an association, it is still a key point to discuss. The sample of this study was mainly composed of parents/caregivers with high schooling when compared to other Brazilian studies, and it is believed that this characteristic can be attributed to the data collection method. It is important to point out that even in a sample with a high level of education, a high prevalence of dental pain was observed. From a public health approach, these findings indicate that considerable attention should be given to oral health promotion and the development of public policies focusing on the population level [11,13,14]. 

The limitations of this study include its cross-sectional design, which does not establish a causal relationship between parental reports of a child’s dental pain experience and exposure. The self-report response bias may also be a limitation. Due to social desirability, parents/caregivers may tend to respond to the PSDQ by indicating a lower frequency, making it difficult to classify their parenting styles accurately. The authors believe that ensuring the confidentiality of responses and conducting data collection through an online questionnaire may have minimized the level of self-reported bias. Moreover, parental reports of children’s dental pain experience might not always accurately reflect the child’s perception. However, they play an important role in planning dental treatment, as the experience of their child’s dental pain is often associated with the status of caries [1,7,8]. Therefore, parental reports of children’s dental pain experience are a valid tool that assists in dental diagnosis, guides treatment decisions, and determines the choice of management techniques.

In the present study, the snowball sampling method was adopted, and the online questionnaires were prepared using Google Forms. Due to the current scenario of the great dissemination of digital tools and technologies, it was decided that the electronic version of this questionnaire would be used since such instruments are reliable and help to make the data collection process more efficient [20,34]. Despite its advantages, it is important to note that this method does not employ random selection. Consequently, our findings should be interpreted with caution and cannot be extrapolated to other populations. Further studies with samples of parents with a good proportion of different parenting styles should be encouraged. The strengths of this study include the original evidence regarding the relationship between children’s dental pain experience and parental styles. Further studies should be conducted to explore the variables that affect children’s behaviors, which can guide the implementation of healthcare policies and preventive dental care measures [11,12]. 

## 5. Conclusions

Based on these results, parents’/caregivers’ attitudes and practices related to dietary habits, such as high sugar consumption by children, can influence children’s oral health. A delay in the first dental appointment may increase the likelihood that children experience dental pain. Thus, in clinical practice, pediatric dentists should encourage parents/caregivers to take their children for their first dental appointment within the first year of life and to control sugar consumption as a preventive measure against oral problems and, consequently, dental pain experienced in childhood. 

## Figures and Tables

**Table 1 ijerph-21-00599-t001:** Characterization of the sample.

Quantitative Variables	Mean (SD)
**Child’s age (years)**	5.45 (±1.16)
**Parent’s/caregiver’s age (years)**	35.0 (±6.10)
**Child’s age at the first dental appointment (years)**	2.26 (±1.31)
**Child’s age at the first consumption of sugar (years)**	1.38 (±0.64)
**Authoritarian parenting style domain**	1.94 (±0.56)
**Permissive parenting style domain**	2.34 (±0.57)
**Authoritative parenting style domain**	4.27 (±0.44)
**Categorical variables**	*n* (%)
**Sex**	
Female	151 (50.3)
Male	149 (49.7)
**Type of school ^1^**	
Public	198 (70.7)
Private	82 (29.3)
**Parent’s/caregiver’s schooling**	
≤12 years	134 (44.7)
>12 years	166 (55.3)
**Family income ^2^**	
≤3 BMMW	153 (52.0)
>3 BMMW	141 (48.0)
**Family composition**	
Only mother	53 (17.7)
Mother and father	247 (82.3)
**First dental visit**	
No	3 (24.3)
Yes	227 (75.7)
**Daily frequency of tooth brushing**	
None or once a day	28 (9.3)
Two or more times a day	272 (90.7)
**Use of fluoridated toothpaste**	
No	5 (8.3)
Yes	275 (91.7)
**Free sugar consumption**	
High	121 (40.3)
Low	179 (59.7)
**Parental reports of children’s dental pain experience**	
No	239 (79.7)
Yes	61 (20.3)
**Predominant parenting style**	
Authoritarian parenting style	2 (0.7)
Permissive parenting style	2 (0.7)
Authoritative parenting style	296 (98.6)

BMMW (Brazilian monthly minimum wage); SD (standard deviation). ^1^ A total of 20 children did not study; ^2^ 6 parents/caregivers did not answer.

**Table 2 ijerph-21-00599-t002:** Characterization of free sugar consumption groups defined by cluster analysis based on item scores of the dietary questionnaire (*n* = 300).

Diet Questionnaire Items	High-Free Sugar Consumption Group	Low-Free Sugar Consumption Sugar	*p*-Value ^1^
	*n* (%)	*n* (%)	
**1. How often does your child drink sugar milk or chocolate milk?**			**<0.001**
Never or rarely	33 (27.3)	83 (46.4)	
1 or 2 times a week	8 (6.6)	27 (15.1)	
3 to 6 times a week	18 (14.9)	29 (16.2)	
1 or 2 times a day	52 (43.0)	36 (20.1)	
3 or more times a day	10 (8.2)	4 (2.2)	
**2. How often does your child drink soft drinks or juice boxes?**			**<0.001**
Never or rarely	0 (0)	72 (40.2)	
1 or 2 times a week	24 (19.8)	74 (41.3)	
3 to 6 times a week	53 (43.8)	29 (16.2)	
1 or 2 times a day	35 (28.9)	4 (2.3)	
3 or more times a day	9 (7.5)	0 (0)	
**3. How often does your child eat candies, lollipops, and chocolate?**			**<0.001**
Never or rarely	7 (5.8)	31 (17.3)	
1 or 2 times a week	27 (22.3)	133 (74.3)	
3 to 6 times a week	66 (54.5)	10 (5.6)	
1 or 2 times a day	17 (14.0)	5 (2.8)	
3 or more times a day	4 (3.4)	0 (0)	
**4. How often does your child eat stuffed cookies?**			**<0.001**
Never or rarely	0 (0)	33 (18.4)	
1 or 2 times a week	22 (18.2)	106 (59.2)	
3 to 6 times a week	52 (43.0)	37 (20.7)	
1 or 2 times a day	39 (32.2)	3 (1.7)	
3 or more times a day	8 (6.6)	0 (0)	

^1^ Chi-square test. Note: Bold values indicate *p* < 0.05.

**Table 3 ijerph-21-00599-t003:** Bivariate analysis of the association between sociodemographic and behavioral variables with parental reports of children’s dental pain experience.

**Covariables**	Parental Reports of Children’s Dental Pain Experience *n* (%)		** *p* ** **-Value ^1^**
	**No**	**Yes**	
**Sex**			
Female	121 (80.1)	30 (19.9)	0.886
Male	118 (79.2)	31 (20.8)	
**Type of school ^2^**			
Public	155 (78.3)	43 (21.7)	0.749
Private	66 (80.5)	16 (19.5)	
**Parent’s/caregiver’s schooling**			
≤12 years	99 (73.9)	35 (26.1)	0.030
>12 years	140 (84.3)	26 (15.7)	
**Family income ^3^**			
≤3 BMMW	116 (75.8)	37 (24.2)	0.080
>3 BMMW	119 (84.4)	22 (15.6)	
**Family composition**			
Only mother	40 (75.5)	13 (24.5)	0.452
Mother and father	199 (80.6)	48 (19.4)	
**Daily frequency of tooth brushing**			
None or once a day	21 (75.0)	7 (25.0)	0.621
Twice or more	218 (80.1)	54 (19.9)	
**Use of fluoridated toothpaste**			
No	20 (80.0)	5 (20.0)	1.000
Yes	219 (79.6)	56 (20.4)	
**Free sugar consumption**			
High	88 (72.7)	33 (27.3)	**0.019**
Low	151 (84.4)	28 (15.6)	

BMMW (Brazilian monthly minimum wage); ^1^ Chi-square test; ^2^ A total of 20 children did not study; ^3^ 6 parents/caregivers did not answer. Note: Bold values indicate *p* < 0.05.

**Table 4 ijerph-21-00599-t004:** Bivariate analysis of the association between quantitative sociodemographic and behavioral variables with the parental reports of children’s dental pain experience.

Variables	Parental Report of Children’s Dental Pain Experience		*p*-Value ^1^
	No Mean (SD)	Yes Mean (SD)	
**Parent’s/caregiver’s age**	5.17 (±6.11)	34.42 (±6.07)	0.475
**Child’s age at the first consumption of sugar**	1.44 (±0.65)	1.15 (±0.55)	**0.001**
**Child’s age at the first dental appointment**	2.08 (±1.21)	2.86 (±1.46)	**<0.001**
**Authoritarian parenting style domain**	1.90 (±0.54)	2.09 (±0.61)	**0.029**
**Permissive parenting style domain**	2.32 (±0.58)	2.41 (±0.55)	0.184
**Authoritative parenting style domain**	4.28 (±0.45)	4.21 (±0.41)	0.188

SD (standard deviation); ^1^ Mann–Whitney test. Note: Bold values indicate *p* < 0.05.

**Table 5 ijerph-21-00599-t005:** Adjusted Poisson regression analyses between sociodemographic and behavioral variables with parental report of children’s dental pain experience.

Covariables	Adjusted PR (95% CI)	*p*-Value
**Free sugar consumption**		
High	1.90 (1.21–3.00)	**0.006**
Low	1	
**Parent’s/caregiver’s schooling**		
≤12 years	1.23 (0.76–1.98)	0.398
>12 years	1	
**Child’s age at the first dental appointment**	1.02 (1.01–1.03)	**<0.001**
**Child’s age at the first consumption of sugar**	0.96 (0.93–1.00)	0.065
**Authoritarian parenting style domain**	1.30 (0.87–1.93)	0.194
**Permissive parenting style domain**	1.10 (0.73–1.67)	0.653
**Authoritative parenting style domain**	1.27 (0.75–2.16)	0.376

RR (relative ratio); CI (confidence interval); BMMW (Brazilian monthly minimum wage). Note: Bold values indicate *p* < 0.05.

## Data Availability

The data presented in this study are available on request from the corresponding author due to privacy concerns regarding the participants and ethical considerations.

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
