# Peer review of "Parental Reports of Children’s Dental Pain Experience and Associated Factors among Brazilian Children"

_ijerph, 2024, doi:10.3390/ijerph21050599_

Round 1

Reviewer 1 Report

Comments and Suggestions for Authors

The authors have interestingly drafted a study to assess the prevalence of parental reports of the child’s dental pain experience and associated factors. However, there are some points of improvement:

1. Citation format should be checked for consistency.

2. There are some queries to the Pilot Study that need to be addressed. Result of the pilot study? Reliability and validity testing of the questionnaire?

3. Include more limitations like Sampling Method, Self-reporting Bias, Recall Bias, Limited Scope of Variables, Cross-sectional Design, Online Survey Limitations, and Potential Confounders.

4. The last statement of the Conclusions should be rephrased to highlight the study on dental pain.

Author Response

On behalf of all authors, I greatly appreciate the comments provided in the review report and believe that the suggested changes have been important in enhancing the quality of the paper. Below, we detail the revisions made to the manuscript and address the comments point by point. All revisions to the manuscript have been accepted and are highlighted in red font.

Sincerely,

Review’s comment: “Citation format should be checked for consistency.”

Author's response: All citations have been checked and standardized.

Review’s comment: “There are some queries to the Pilot Study that need to be addressed. Result of the pilot study? Reliability and validity testing of the questionnaire?”

Author's response: The pilot study tested the applicability of the methods and involved ten parents/caregivers. Participants were selected for convenience and were not included in the main study sample. No changes to the methodology were deemed necessary. In the main study sample, we assessed the reliability of the short Brazilian version of the Parenting Styles and Dimensions Questionnaire (PSDQ) using Cronbach's alpha index, obtaining a satisfactory value of 0.74. We have added this information to the Questionnaire section, as follows: “Cronbach’s alpha index of the instrument was evaluated, obtaining a satisfactory value of 0.74.” [Page 4, Line 154]

Review’s comment: “Include more limitations like Sampling Method, Self-reporting Bias, Recall Bias, Limited Scope of Variables, Cross-sectional Design, Online Survey Limitations, and Potential Confounders.”

Author's response: We appreciate your suggestion and have included more limitations to the text, as follows: The limitations of this study include its cross-sectional design, which does not establish a causal relationship between parental report of child’s dental pain experience and the exposures. The self-report response bias may be also a limitation. Due to social desirability, parents/caregivers may tend to respond to the PSDQ by indicating a lower frequency, making it difficult to classify their parenting styles accurately. The authors believe that ensuring the confidentiality of responses and conducting data collection through an online questionnaire may have minimized the level of self-reported bias. Moreover, parental report of child's dental pain experience might not always accurately reflect the child's perception. However, they play an important role in planning dental treatment, as the experience of child’s dental pain is often associated with caries status. Therefore, parental report of child’s dental pain experience is a valid tool that assists in dental diagnosis, guides treatment decisions, and determines the choice of management techniques. [Page 9, line 278]

            Despite its advantages, it's important to note that this method does not employ random selection, thus it is not representative of the entire population. Consequently, the findings are limited to the studied sample. [Page 9, line 294]

Review’s comment: “The last statement of the Conclusions should be rephrased to highlight the study on dental pain.”

Author's response: We appreciate your suggestion and have reformulated the last statement, as follows: Thus, in clinical practice, pediatric dentists should encourage parents/caregivers to take their children for their first dental appointment within the first year of life and to control sugar consumption as a preventive measure against oral problems and, consequently, dental pain experience in childhood. [Page 10, Line 305]

Reviewer 2 Report

Comments and Suggestions for Authors

The topic of the manuscript is interesting and suitable for publishing in IJERPH journal.

Page 2, Study design: ”Divinópolis, which is located in the 90 state of Minas Gerais in the southeast region of Brazil”. Please add more details, to get the context of the study population.

Page 2, line 93: ”children with neurological disorders, such autism and epilepsy, in addition to systemic diseases, such as diabetes, were excluded” – perhaps it is better to write parents with children with .......were excluded.

Line 101: 10 schools...Please detail about the child age of starting school, etc. How many schools are in the study area?

Lines 127-129. This sentence is not clear. Parents/caregivers were asked to answer these items based on a six-month recall according to the response options consisting of “never or rarely”, “one or two times a week”, “three to six times a week”, “one or two times a day” and “three or more times a day”.

Line 149: why did the authors mention ”10 children and their parents/caregiver”. As far as I undesrstand, in the study participated only the parents.

Lines 171-175: The same mention as above. Why did you mention children?? ”Seven children were excluded for not 172 being in the age group, six children were excluded for having autism, one child was ex- 173 cluded for having epilepsy, and another one for having acute disseminated encephalomy- 174 elitis. Thus, the final sample consisted of 300 parents/caregivers and their children”.

Results section: In my opinion, table 2 should be presented first, as Table 1.

Kind regards,

Author Response

On behalf of all authors, I greatly appreciate the comments provided in the review report and believe that the suggested changes have been important in enhancing the quality of the paper. Below, we detail the revisions made to the manuscript and address the comments point by point. All revisions to the manuscript have been accepted and are highlighted in red font.

Sincerely,

Review’s comment: “The topic of the manuscript is interesting and suitable for publishing in the IJERPH journal.”

Author's response: We appreciate your dedication to this review and the care you took in conducting it.

Review’s comment: “Page 2, Study design: ”Divinópolis, which is located in the state of Minas Gerais in the southeast region of Brazil”. Please add more details, to get the context of the study population.”

Author's response: We provided data on the Gini index of Divinópolis and the number of private and public preschools and elementary schools catering to children within the specified age group to enhance the population description, as follows: The city has a Gini Index of 0,47. The study sample consisted of parents/caregivers of children aged four to seven years, of both sexes, living in Divinópolis. The study was conducted at public and private preschools and elementary schools. Preschool covers the age range from 4 to 5 years old. There are 74 preschools in Divinópolis (37 public and 37 private). Elementary education includes children aged six to fourteen, and the city has 73 schools (51 public and 22 private). [From page 2, Line 91 to Page 3, Line 97]

Review’s comment: “Page 2, line 93: ”children with neurological disorders, such autism and epilepsy, in addition to systemic diseases, such as diabetes, were excluded” – perhaps it is better to write parents with children with .......were excluded.”

Author's response: Given that our sample consisted of parents and caregivers rather than children, we understand your concern. To clarify any potential misunderstandings, we have revised the sentence, as follows: Parents/caregivers of children with neurological disorders, such as autism and epilepsy, in addition to systemic diseases, such as diabetes, were excluded. [Page 3, Line 104]

Review’s comment: “Line 101: 10 schools...Please detail about the child age of starting school, etc. How many schools are in the study area?”

Author’s response: We detailed the number of schools in the study area, as follows: The study was conducted at public and private preschools and elementary schools. Preschool covers the age range from 4 to 5 years old. There are 74 preschools in Divinópolis (37 public and 37 private). Elementary education includes children aged six to fourteen, and the city has 73 schools (51 public and 22 private). [Page 2, Line 93]

Review’s comment: “Lines 127-129. This sentence is not clear. Parents/caregivers were asked to answer these items based on a six-month recall according to the response options consisting of “never or rarely”, “one or two times a week”, “three to six times a week”, “one or two times a day” and “three or more times a day”.”

Author's response: We rewrote the sentence to improve understanding, as follows: Parents/caregivers were asked to respond to these items based on a six-month recall. The response options included "never or rarely," "once or twice a week," "three to six times a week," "once or twice a day," and "three or more times a day." [Page 3, Line 134]

Review’s comment: “Line 149: why did the authors mention ”10 children and their parents/caregiver”. As far as I understand, in the study participated only the parents.”

Author's response: We appreciate your correction. We revised the sentence, as follows: A pilot study was carried out with ten parents/caregivers to test the applicability of the proposed methodology. [Page 4, Line 157]

Review’s comment: “Lines 171-175: The same mention as above. Why did you mention children?? ”Seven children were excluded for not being in the age group, six children were excluded for having autism, one child was excluded for having epilepsy, and another one for having acute disseminated encephalomyelitis. Thus, the final sample consisted of 300 parents/caregivers and their children”.”

Author's response: Again, we appreciate your correction. We revised the sentence, as follows: Seven parents/caregivers were excluded because their children did not meet the age group criteria, and eight were excluded due to their children being diagnosed with autism, epilepsy, or acute disseminated encephalomyelitis. Thus, the final sample consisted of 300 parents/caregivers. [Page 4, Line 181]

Review’s comment: “Results section: In my opinion, table 2 should be presented first, as Table 1.”

Author's response: Thank you for your suggestion. We have reorganized the tables, resulting in changes to the table titles and the text in the results section. [Page 4, Line 192]

Reviewer 3 Report

Comments and Suggestions for Authors

There must be institutional ethical committee approval of the study besides consent of the parents 

It would be more interesting if the questionnaire data compared with the caries status of the children (dmft) 

There should be more explanation of the sample size design (why 10 schools why 300 samples)

The hypothesis on parenting styles and children dental pain experiences was not valid (ln 226-227 and 275 - 276) due to the characteristic of distributing samples (Table 2) showed 98.6% Authoritative parenting style. therefore the analysis of this factor was impossible in Table 4 & 5

Comments on the Quality of English Language

Author Response

On behalf of all authors, I greatly appreciate the comments provided in the review report and believe that the suggested changes have been important in enhancing the quality of the paper. Below, we detail the revisions made to the manuscript and address the comments point by point. All revisions to the manuscript have been accepted and are highlighted in red font.

Sincerely,

Review’s comment: “There must be institutional ethical committee approval of the study besides the consent of the parents.”

Author's response: We appreciate your comment. According to the IJERPH guidelines, the institutional ethical committee approval for the study should be stated by the authors in the section "Institutional Review Board Statement" before the references. Thus, we provide the following statement only in this section to avoid duplicated information: “The study was conducted following the Declaration of Helsinki and was approved by the Research Ethics Committee of XXXX, Brazil (protocol #3.589.079).” [Page 10, Line 314]

Review’s comment: ”It would be more interesting if the questionnaire data compared with the caries status of the children (dmft).”

Author's response: While we agree that incorporating clinical data on dental caries experience via the dmft index could enhance the robustness of this study, it's important to note that the literature indicates pain as a symptom commonly linked with untreated dental caries lesions (Clementino et al 2015; Paredes et al 2021). Pain assessment plays an important role in dental treatment planning, aiding in dental diagnosis, treatment decisions, and the selection of management techniques. We acknowledge that parental reports of child’s dental pain experience might not always accurately reflect children's perceptions. Therefore, we include the following sentence in the discussion: Moreover, parental report of child's dental pain experience might not always accurately reflect the child's perception. However, they play an important role in planning dental treatment, as the experience of child’s dental pain is often associated with caries status. Therefore, parental report of child’s dental pain experience is a valid tool that assists in dental diagnosis, guides treatment decisions, and determines the choice of management techniques. [Page 9, Line 284]

Review’s comment: “There should be more explanation of the sample size design (why 10 schools why 300 samples)”

Author's response: We added more explanation of the sample size design, as follows: The study population was recruited using snowball sampling [21]. In this method, the first subject is recruited, and then he/she provides multiple referrals. Each new referral then provides more data for referral and so on, until saturation. In this study, the saturation point was determined based on the data collection period (between August and November 2021). The first referrals were parents/caregivers of children from 10 schools (five public and five private) selected for convenience. Additionally, a group of health professionals (pediatric dentist, pediatrician, nutritionist, speech therapist, and psychologist) and parents/caregivers were invited to assist in the questionnaire dissemination. Parents/caregivers of children with neurological disorders, such as autism and epilepsy, in addition to systemic diseases, such as diabetes, were excluded. [Page 2, line 97]

            In addition, we inserted a limitation of this kind of sample selection in the discussion, as follows: Despite its advantages, it's important to note that this method does not employ ran-dom selection, thus it is not representative of the entire population. Consequently, the findings are limited to the studied sample. [Page 9, line 294]

Review’s comment: “The hypothesis on parenting styles and children dental pain experiences was not valid (ln 226-227 and 275 - 276) due to the characteristic of distributing samples (Table 2) showed 98.6% Authoritative parenting style. Therefore the analysis of this factor was impossible in Table 4 & 5.”

Author's response: We believe that the hypothesis regarding parenting styles and children's experience of dental pain is valid. The short Brazilian version of the Parenting Styles and Dimensions Questionnaire assessed parents'/caregivers' parenting styles through 32 questions, distributed across three domains: authoritative, authoritarian, and permissive. The arithmetic mean of the total score for each domain was calculated, ranging from 1 to 5. The highest score among the three domains indicated the predominant parenting style (Table 2, after revision, is Table 1). For instance, consider a parent/caregiver with the following total score averages: 4.47 (indicating dominance of the authoritative parenting style), 1.92 (indicating dominance of the authoritarian parenting style), and 2.20 (indicating dominance of the permissive parenting style). This parent/caregiver would be classified as predominantly authoritative. Thus, Table 1 shows the frequency of predominant styles in the sample, with the authoritative parenting style being the most prevalent. However, it's important to emphasize that originally parenting styles were conceptualized as non-mutually exclusive typologies. This means that a higher score in a specific parenting style indicates a greater frequency of parenting practices related to that style reported by the parent/caregiver. Consequently, the parenting style was incorporated into the analysis as mean scores (quantitative variables) of the three domains of parenting styles (Tables 4 and 5). This approach is valid because, even though a parent/caregiver may have a dominant parenting style, they may exhibit varying degrees of attitudes related to each of the three domains.

            To clarify the plausibility of using the parental style variable in its quantitative form, we rewrite the following sentence in the statistical analysis section: “In the present study, the parenting style was incorporated into the analysis through the mean scores (quantitative variables) of the three domains of parenting styles. This analysis is valid because parenting styles were conceptualized as non-mutually exclusive typologies. Thus, although a parent/caregiver has a dominant parenting style, he may present more or fewer attitudes related to each of the three domains.” [Page 4, Line 169]

Round 2

Reviewer 2 Report

Comments and Suggestions for Authors

The authors addressed all the comments. 

Author Response

On behalf of all the authors, I appreciate the opportunity to address the points you raised. Thank you for your time and dedication to our paper.

Sincerely,

Reviewer 3 Report

Comments and Suggestions for Authors

Even though most reviewer's points been revised but the topic on patenting styles not yet resolved. 

The result showed 98.6% Authoritative parenting style which meant the poor distribution of the data of the 3 groups of parenting styles . Therefore the analysis of this factor was impossible in Table 4 & 5. Suggested that this facor should be mentioned that due to the undistribution of the data in this factor therefore it could not express the correlation of this parenting style in this study more sample size or more data collected for the even distribution will make hte analysis of this factor possible in the future.

Author Response

I appreciate the opportunity to clarify the not fully explained point further. Below, we respond to your comment. Revisions have been highlighted in red font in the manuscript.

Sincerely,

Review’s comment: “Even though most reviewer's points been revised but the topic on patenting styles not yet resolved. The result showed 98.6% Authoritative parenting style which meant the poor distribution of the data of the 3 groups of parenting styles. Therefore the analysis of this factor was impossible in Table 4 & 5. Suggested that this facor should be mentioned that due to the undistribution of the data in this factor therefore it could not express the correlation of this parenting style in this study more sample size or more data collected for the even distribution will make hte analysis of this factor possible in the future.”

Author's response: The aim of our study was not to assess the influence of a specific parenting style predominance in the outcome but rather to assess the influence of the degree of each parenting style (more or less authoritative, more or less authoritarian, and more or less permissive) in parental reports of child's dental pain experience. Therefore, we evaluated the mean scores of each parenting style. This approach is justified as parenting styles were originally conceptualized as non-mutually exclusive typologies. This implies that every parent or caregiver exhibits all three parenting styles to a greater or lesser degree.

            To clarify your point, we rewrote the last paragraph of the ´Introduction´ section, stating: “The present study aimed to assess the prevalence of parental report of the child’s dental pain experience and associated factors, such as sociodemographic factors, parenting styles (more or less authoritative, more or less authoritarian, and more or less permissive), free sugar consumption, and oral hygiene practices in Brazilian children.” [Page 2, Line 82]

            We present the results of the frequencies of the predominant parenting styles in Table 1, as you mentioned the authoritative parenting style being the majority in our sample (98.6%). Additionally, in Table 1, we present the mean scores and the respective standard deviations of the three parenting styles. In Tables 4 and 5, we used the mean scores of each parenting style in the analysis.

            Other points should also be addressed:

  1. The hypothesis that parental styles could be related to the child’s dental pain experience was not confirmed in our study. But, for instance, consider a hypothetic result in the Poisson regression model (mean score of authoritative parental style domain x child's dental pain experience): RR = 0.37; 95% CI = 0.16 – 0.86; P = 0.022. This finding could be interpreted as follows: an increase of one point in the authoritative parental style mean score resulted in a 63% decrease in the report of the child's experience of dental pain, or more authoritative parents/caregivers reported a lower prevalence of the child's dental pain experience.
  2. The high percentage of authoritative parents/caregivers may indicate the normal distribution of this study’s sample, as also previously reported in the literature. However, self-report response bias must be considered due to social desirability, which reflects individuals' tendency to respond according to social norms and expectations. This bias may have occurred in items related to physical punishment in the authoritarian parenting style domain (e.g., “I grab our child when he is being naughty”). Parents/caregivers may tend to respond to these items by indicating a lower frequency, making it difficult to classify them as authoritarian parents. We acknowledged this limitation in the discussion, stating: “The self-report response bias may also be a limitation. Due to social desirability, parents/caregivers may tend to respond to the PSDQ by indicating a lower frequency, making it difficult to accurately classify their parenting styles. The authors believe that ensuring the confidentiality of responses and conducting data collection through an online questionnaire may have minimized the level of self-reported bias.” [Page 9, Line 278]
  3. The explanation provided above regarding the difficulty in identifying parents/caregivers as authoritarian or permissive also justifies the need for using the means of each parental style score in analysis.
  4. We used a non-probabilistic method to select our sample. Thus, the results of our study should be interpreted with caution and cannot be extrapolated to other populations. We acknowledged this limitation in the discussion, stating: “Despite its advantages, it's important to note that this method does not employ random selection. Consequently, our findings should be interpreted with caution and cannot be extrapolated to other populations. Further studies with samples of parents with a good proportion of different parenting styles should be encouraged.” [Page 9, Line 294]
